# Development and Validation of an Eating-Related Eco-Concern Questionnaire

**DOI:** 10.3390/nu14214517

**Published:** 2022-10-27

**Authors:** Baiyu Qi, Emily K. Presseller, Gabrielle E. Cooper, Avantika Kapadia, Alexis S. Dumain, Shantal M. Jayawickreme, Emily C. Bulik-Sullivan, Eric F. van Furth, Laura M. Thornton, Cynthia M. Bulik, Melissa A. Munn-Chernoff

**Affiliations:** 1Department of Epidemiology, University of North Carolina at Chapel Hill, Chapel Hill, NC 27516, USA; 2Department of Psychological and Brain Sciences, Drexel University, Philadelphia, PA 19104, USA; 3Center for Weight, Eating, and Lifestyle Science, Drexel University, Philadelphia, PA 19104, USA; 4Department of Psychiatry, University of North Carolina at Chapel Hill, Chapel Hill, NC 27599, USA; 5Department of Clinical Psychology, National Institute of Mental Health and Neurosciences, Bangalore 560030, India; 6Department of Psychology and Neuroscience, University of North Carolina at Chapel Hill, Chapel Hill, NC 27599, USA; 7Marsico Lung Institute and Cystic Fibrosis Research Center, University of North Carolina at Chapel Hill, Chapel Hill, NC 27599, USA; 8Department of Cell Biology and Physiology, University of North Carolina at Chapel Hill, Chapel Hill, NC 27599, USA; 9GGZ Rivierduinen Eating Disorders Ursula, 2333 ZZ Leiden, The Netherlands; 10Department of Psychiatry, Leiden University Medical Center, 2333 ZB Leiden, The Netherlands; 11Department of Medical Epidemiology and Biostatistics, Karolinska Institutet, 171 77 Stockholm, Sweden; 12Department of Nutrition, University of North Carolina at Chapel Hill, Chapel Hill, NC 27516, USA

**Keywords:** eco-anxiety, eating behaviors, climate change, questionnaire development, mental health

## Abstract

Eco-concern, the distress experienced relating to climate change, is associated with mental health, yet no study has examined disordered eating related to eco-concern. This study developed and validated a 10-item scale assessing Eating-Related Eco-Concern (EREC). Participants (*n* = 224) completed the EREC, Climate Change Worry Scale (CCWS), and Eating Disorder Examination-Questionnaire (EDE-Q). Construct validity, convergent validity, and internal consistency were evaluated. Sex differences in EREC were evaluated using *t*-tests. Associations among the EREC, CCWS, and EDE-Q were evaluated using linear regression models. Sensitivity analyses were conducted in individuals below EDE-Q global score clinical cut-offs. Factor analysis suggested that all items loaded adequately onto one factor. Pearson’s correlation and Bland–Altman analyses suggested strong correlation and acceptable agreement between the EREC and CCWS (*r* = 0.57), but weak correlation and low agreement with the EDE-Q global score (*r* = 0.14). The EREC had acceptable internal consistency (*α* = 0.88). No sex difference was observed in the EREC in the full sample; females had a significantly higher mean score than males in sensitivity analysis. The EREC was significantly positively associated with the CCWS and EDE-Q global and shape concern scores, but not in sensitivity analysis. The EREC is a brief, validated scale that can be useful to screen for eating-related eco-concern.

## 1. Introduction

Climate change is defined as alterations in weather patterns and temperatures caused by both natural disasters and human activity [1]. The world has encountered progressively worsening climate change, which is known to have a significant impact on food insecurity, housing, and physical health [2]. More recently, the adverse impact of climate change on mental health has been documented, with 25−50% percent of individuals who are exposed to extreme climate-related events subsequently developing psychiatric conditions [3]. As a result, the more general emotional distress induced by climate change, known as eco-concern, is rapidly gaining attention. Prior studies have indicated an association between eco-concern and psychopathology; however, the connection between eco-concern and disordered eating (e.g., dietary restraint, eating concern, weight concern, and shape concern) remains understudied [3,4]. Thus, the current study explored this potential relationship, proposing that an association may exist between eco-concern and disordered eating.

Eco-concern is a general term for the distress one experiences specifically related to the climate crisis; it may include emotional disturbance due to environmental changes, symptoms of anxiety and distress (e.g., rumination), helplessness, dread, guilt, sadness, doom, and frustration [5,6]. Eco-concern may also be associated with specific coping and affect regulation strategies, such as cognitive reframing of threats and altering consumer habits, as well as other mental health conditions, such as anxiety, depression, substance use disorders, and post-traumatic stress disorder [3,7,8,9,10].

Numerous factors may influence the association between eco-concern and mental health outcomes. Age may be an important factor: eco-concern is known to impact children, adolescents, and young adults the most, particularly those with preexisting mental health conditions and lacking social support [4,11,12]. In addition, constant media exposure has exacerbated eco-concern [13]. Relatedly, the recently examined phenomenon of “doomscrolling”—the act of compulsively scrolling through distressing information—may worsen anxiety about climate change [14]. Moreover, individuals who are more invested in and aware of climate change and sustainability and those who live in areas more prone to extreme weather events and natural disasters are also at higher risk of experiencing worsening mental health outcomes [15,16]. Lastly, feelings of diminished control, along with uncertainty and powerlessness regarding one’s future related to climate change forecasts, may contribute to the association between eco-concern and adverse mental health outcomes [11,17,18].

The manner in which eco-concern impacts disordered eating and the magnitude of this effect are unknown. However, studies have examined associations between specific eating behaviors related to climate change. Undergraduate students, particularly female students, may adopt behaviors such as eliminating meat, seeking organic and/or local foods, and avoiding food waste in response to climate change concerns [19]. Although such behaviors are not harmful themselves, and are, in fact, beneficial for the environment, when taken to the extreme, they could represent more concerning behavioral patterns (e.g., elimination of entire food groups, rigidity of eating choices) similar to established patterns of disordered eating (e.g., dietary restraint). A survey of 2000 children and teens in the UK found that 17% reported that climate concerns impact their sleeping and eating behaviors [20]. However, no study to our knowledge has comprehensively explored the specific association of concerns about climate change with eating behaviors in the general population and whether those eating behaviors reflect disordered eating. Emerging from clinical observations of individuals with eating disorders (e.g., anorexia nervosa, bulimia nervosa, and binge-eating disorder) who endorse disordered eating related to concerns about climate change, the goal of this study was to provide insight into specific changes to eating behavior prompted by eco-concern and its relationship with disordered eating in a general population sample.

The current study addressed research gaps by: (1) developing and validating the first questionnaire assessing eating behaviors related to eco-concern, the eating-related eco-concern (EREC) scale; (2) examining sex differences in climate change worry and eating-related eco-concern; and (3) examining associations among climate change worry, eating-related eco-concern, and disordered eating. In this initial study, there were three hypotheses: (1) the EREC scale would have good psychometric properties (construct validity, convergent validity, and internal consistency) in this sample; (2) female participants would score higher on both climate change worry and eating-related eco-concern; and (3) eating-related eco-concern would be associated with both climate change worry and disordered eating. This study provides the first questionnaire to screen for eating behaviors related to eco-concern and brings awareness to eco-concern as a potentially important contributing factor in the development of disordered eating.

## 2. Materials and Methods

### 2.1. Participants

Participants were recruited from across the United States via flyers and online recruitment strategies, including posting on social media, ResearchMatch, and University of North Carolina at Chapel Hill listservs. The only inclusion criterion for the present study was that individuals were at least 18 years old; there were no exclusion criteria. Participants provided online consent and then completed all questionnaires via an online survey. Participation was anonymous. A total of 257 adults took the survey. We excluded participants who had incorrect answers to quality assessment questions (*n* = 6), those who had missing data on eating-related eco-concern (*n* = 24), and those with inflammatory responses (*n* = 3), resulting in a final sample size of 224. Compared with participants who were excluded (*n* = 33), the analytical sample had a higher proportion of White/Caucasian participants (82% vs. 50%), but did not differ in age or sex assigned at birth. A total of 31 participants had missing data on any EDE-Q subscales, and thus the global score. Compared with these participants, those without missing data on EDE-Q scores were significantly older (*Mean* = 39.01, *SD* = 16.97 vs. *Mean* = 27.30 years, *SD* = 11.59), more likely to be male (22% vs. 13%), and had higher current BMI (*Mean* = 25.79, *SD* = 5.74 vs. *Mean* = 20.87, *SD* = 4.56).

This study was reviewed and approved by The University of North Carolina at Chapel Hill Biomedical Institutional Review Board. Due to the anonymous nature of the questionnaires, the committee determined this study to be exempt from further review.

### 2.2. Measures

#### 2.2.1. Demographic Information

Participants self-reported demographic information, including age, sex assigned at birth, gender, race, and ethnicity. Participants self-reported their current height and weight from which body mass index was calculated.

#### 2.2.2. Climate Change Worry Scale

The Climate Change Worry Scale (CCWS) is a 10-item scale assessing worry about climate change [21]. Items were scored on a *1* = *Never* to *5* = *Always* Likert scale. The score was computed as the sum of all 10 items. The scale has demonstrated good internal consistency (Cronbach’s *α* = 0.95), factor structure invariance, and test–retest reliability (*r* = 0.91) [21]. The CCWS also demonstrates convergent and divergent validity with widely used clinical measures of worry, anxiety, and weather-related fear [21]. In this study, it had an internal consistency of 0.93.

#### 2.2.3. Eating-Related Eco-Concern Scale Item Development

We developed the Eating-Related Eco-Concern (EREC) scale, which is a 10-item assessment regarding the degree to which individuals consider ecological impact when making food choices due to concerns about the changing climate. The items were based on our clinical observations, the previously published CCWS [21], and a literature review on eco-friendly eating and sustainable eating. Items were scored on a 5-point Likert scale with the anchors *1* = “*Never*”, *2* = “*Rarely*”, *3* = “*Sometimes*”, *4* = “*Often*”, and *5* = “*Always*”. The score was calculated by summing all items, which ranged from 10 to 50. The full questionnaire is included in the Appendix A.

#### 2.2.4. Eating Disorder Examination-Questionnaire

The Eating Disorder Examination-Questionnaire (EDE-Q) 6.0 is a 28-item self-report questionnaire designed to measure disordered eating over the last 28 days [22]. Twenty-two items were included, measuring four subscales: dietary restraint (5 items; measures restraint over eating, avoidance of eating, and dietary avoidance), eating concern (5 items; measures the preoccupation with food, eating in secret, and guilt about eating), shape concern (8 items; measures the desire for a flat stomach, the importance of body shape, and fear of weight gain), and weight concern (5 items; measures the importance of weight, dissatisfaction with weight, and the desire to lose weight). Responses were on a 7-point rating scale (0–6). Subscale scores were calculated by taking the mean of all items in each subscale, with higher scores indicating a greater degree of disordered eating. The global score was calculated as the mean of the four subscales scores. The EDE-Q has been validated in non-clinical [23,24,25,26] and clinical eating disorder samples [27,28]. The EDE-Q demonstrates high internal consistency [24,25,29], test–retest reliability [25,28], and good discriminant validity [30,31]. In this study, the Cronbach’s alpha was 0.95 for the global score; 0.82 for restraint; 0.87 for eating concern; 0.91 for shape concern; and 0.81 for weight concern.

### 2.3. Statistical Analyses

All statistical analyses were conducted in SAS version 9.4 [32]. Results with *p* < 0.05 were considered significant. The normality of the EREC score, the CCWS score, and the EDE-Q global and subscale scores were examined by plotting the histograms (Appendix A). The EREC and CCWS scores were normally distributed. The EDE-Q global and subscale scores were all right skewed. We did not conduct transformations for the EDE-Q scores because the inclusion criteria for the sensitivity analysis were based on the EDE-Q global score. Furthermore, it facilitated interpretation of results.

#### 2.3.1. Psychometric Properties of the Eating-Related Eco-Concern Scale

To examine the factor structure of the EREC scale, polychoric correlations were calculated and tested using Bartlett’s test of sphericity to ensure they did not constitute an identity matrix. Results of Bartlett’s test of sphericity were not significant. Factor analysis was conducted with an unweighted least squares factor extraction procedure [33]. The number of factors extracted was determined using multiple approaches, including conducting a parallel analysis and evaluating factor analytic results using Kaiser’s Rule (i.e., eigenvalues > 1), the scree plot, and factor loading interpretations [34,35]. Factor loadings ≥ 0.40 indicated that the item loaded adequately onto the factor [36,37]. Items with factor loadings < 0.40 were removed from subsequent analyses.

Pearson correlation coefficients were calculated between the EREC score with the CCWS score, and between the EREC score with the EDE-Q global score to examine the strength of the relationship between two scales. Correlations of 0.10 were considered small, 0.30 were medium, and 0.50 or higher were large [38].

Bland–Altman analysis was conducted to examine the level of agreement between the EREC with the CCWS scores, and between the percentages of the EREC score and the EDE-Q global score. Due to different ranges of the EREC score (range 10–50) and EDE-Q global score (range 0–6), these two scores were converted to percentages using the following equations:1 percentage value on the EREC score: (1/50) × 100 = 21 percentage value on the EDE-Q global score: (1/6) × 100 = 16.67

Thus, for each individual, the EREC percentage was calculated as 2 × the EREC score, and the EDE-Q percentage was calculated as 16.67 × the EDE-Q global score.

Bland–Altman plots were visualized, where the X axis was the average of two measures, and the Y axis was the difference between two measures. For each comparison, the mean difference and standard deviation were calculated, and 95% limits of agreement (LOA) were calculated using the equation (Mean difference ± 1.96 * standard deviation). Higher mean differences and wider LOA indicate lower agreement between two measures. The Bland–Altman index, defined as percentage of the difference between two measure falling beyond the LOA, was also calculated for each comparison. A Bland–Altman index below 5% suggests good agreement between two measures [39].

Internal consistency, as measured by Cronbach’s alpha, was evaluated using all items of the EREC scale. An alpha ≥ 0.80 was considered evidence of adequate internal consistency [40].

#### 2.3.2. Differences in Climate Change Worry and Eating-Related Eco-Concern by Sex Assigned at Birth

The differences in the CCWS score and the EREC score between male and female (sex assigned at birth) participants was evaluated using independent samples *t*-tests.

#### 2.3.3. Associations among Climate Change Worry, Eating-Related Eco-Concern, and Disordered Eating

Multiple linear regression models were conducted to examine associations between: (1) climate change worry (predictor) with eating-related eco-concern (outcome); (2) EDE-Q global score and each subscale score (i.e., restraint, eating concern, shape concern, weight concern; predictors) with climate change worry (outcome); and (3) EDE-Q global score and each subscale score (predictors) with eating-related eco-concern (outcome). Age was included as a covariate in all models.

#### 2.3.4. Sensitivity Analysis among Participants with EDE-Q Global Score below Clinical Cut-Off Values

All analyses were replicated in a subset of the sample, which included participants whose EDE-Q global scores were below clinical cut-off values, to examine whether results differ in individuals without a potential eating disorder. A cut-off value of 1.68 was used for males [41], whereas 4.0 was used for females [23,42]. Seven male participants and eight female participants met the respective clinical thresholds, and 31 participants had missing data on EDE-Q global scores. Thus, a total of 178 participants were included in the sensitivity analysis.

## 3. Results

### 3.1. Participant Characteristics

The sample (*N* = 224) had a mean age of 37.43 years (*SD* = 16.78, range 18–89), and 79% (*n* = 177) of the sample reported being assigned female sex at birth. No participant reported being intersex at birth. Participants identified their gender as 77% (*n* = 173) women, 20% (*n* = 44) men, and 3% (*n* = 7) gender non-conforming, gender fluid, questioning or unsure, or other. The racial composition was 82% (*n* = 184) White or Caucasian, 3% (*n* = 6) Black or African American, 11% (*n* = 24) Asian, and 4% (*n* = 10) more than one race or other. The majority (94%; *n* = 210) of the sample was non-Hispanic. Among participants who reported their current height and weight (*n* = 175), mean current body mass index was 25.62 kg/m^2^ (*SD* = 5.76, range = 13.81–45.70). Table 1 summarizes the descriptive statistics for climate change worry, eating-related eco-concern, and disordered eating.

### 3.2. Psychometric Properties of the Eating-Related Eco-Concern Scale

#### 3.2.1. Factor Structure

Table 2 summarizes the mean score (standard deviation) and factor loading for each item of the EREC scale. Results of the factor analysis (including the parallel analysis, Kaiser’s rule, and scree plot) indicated that the EREC scale was comprised of a single factor. All items loaded adequately (factor loading ≥ 0.40) onto the single factor.

#### 3.2.2. Convergent Validity

The EREC score had a large and significant correlation with the CCWS score (*r* = 0.57, *p* < 0.0001), but a weak correlation of 0.14 with the EDE-Q global score (*p* = 0.0455). Bland–Altman analyses demonstrated the agreement between the EREC scale with the CCWS and EDE-Q global score (Figure 1). On average, participants scored 4.84 units lower on the EREC scale than the CCWS, with a moderate 95% LOA of (−19.81, 10.14). Eight out of 224 participants had a difference between the EREC and CCWS scores falling beyond the LOA, resulting in a Bland–Altman index of 3.57%, suggesting a good agreement between these two measures. For the EREC and EDE-Q percentages, the mean difference was 24.75, with a wide LOA of (−21.12, 70.62). The Bland–Altman index was 6.22% (12/193), suggesting that the EREC scale had a low agreement with the EDE-Q global score and the EREC scale captured a distinct construct from disordered eating.

#### 3.2.3. Internal Consistency

The EREC scale demonstrated good internal consistency: Cronbach’s alpha = 0.88.

### 3.3. Examining Differences in Climate Change Worry and Eating-Related Eco-Concern by Sex at Birth

A significant difference was observed in climate change worry between female and male participants (*t* (221) = −2.90, *p* = 0.0042), where female participants demonstrated a significantly higher level of climate change worry (*Mean* = 30.53, *SD* = 8.06) compared with males (*Mean* = 26.57, *SD* = 9.06). Mean scores on eating-related eco-concerns did not differ significantly by sex at birth (*t* (221) = −1.73, *p* = 0.0852, *Mean_females_* = 25.31, *SD_females_* = 7.84, *Mean_males_* = 23.02, *SD_males_* = 8.49).

### 3.4. Associations among Climate Change Worry, Eating-Related Eco-Concern, and Disordered Eating

Table 3 shows the results from the linear regression models. Climate change worry was positively significantly associated with eating-related eco-concern after adjusting for age, indicating that participants with higher CCWS scores also had higher EREC scores. Climate change worry was not significantly associated with any of the examined disordered eating characteristics (i.e., global score, restraint, eating concern, weight concern, and shape concern). Eating-related eco-concern was positively significantly associated with EDE-Q global score and shape concern after adjusting for age. Eating-related eco-concern was not significantly associated with restraint, eating concern, or weight concern. Figure 2 presents the regression between (1) the CCWS and the EREC scores; (2) the EDE-Q global score and the EREC score; and (3) the EDE-Q shape concern score and the EREC score.

### 3.5. Sensitivity Analysis in Participants Whose EDE-Q Global Scores Were below Clinical Cut-Off Values

All analysis were replicated only in participants who did not meet the clinical cut-off values for the EDE-Q global score (*n* = 178). Compared with the whole sample, this subset did not differ significantly in age (*Mean* = 37.43 years, *SD* = 16.78 in the whole sample vs. *Mean* = 39.04 years, *SD* = 16.90 in the subset), sex at birth (79% vs. 81% female), gender identity (77% vs. 80% women), race (82% vs. 84% White or Caucasian), ethnicity (94% vs. 94% non-Hispanic), and current BMI (*Mean* = 25.62 kg/m^2^, *SD* = 5.76 vs. *Mean* = 25.67 kg/m^2^, *SD* = 5.71). The CCWS score, EREC score, EDE-Q global and subscale scores did not differ significantly between the whole sample and the subset used in the sensitivity analysis (Appendix A).

The psychometric properties of the EREC scale remain consistent in the sensitivity analysis. Results from the factor analysis still support a single-factor structure (Appendix A). Pearson’s correlation analysis demonstrates large correlation between the EREC and CCWS scores (*r* = 0.53, *p* < 0.0001), but weak correlation between the EREC score and the EDE-Q global score (*r* = 0.09, *p* = 0.2282). From the Bland–Altman analysis, the EREC scale had a better agreement with the CCWS, but a low agreement with the EDE-Q global score (Appendix A). Cronbach’s alpha was 0.87 in the subset, suggesting good internal consistency.

Female participants demonstrated a significantly higher level of climate change worry (*Mean* = 30.26, *SD* = 7.72) compared with males [*Mean* = 25.94, *SD* = 8.75; *t* (176) = −2.86, *p* = 0.0047] in the subset. In contrast to the results for the whole sample, female participants in the subset had higher levels of eating-related eco-concern than male participants (*t* (176) = −2.13, *p* = 0.0348, *Mean_females_* = 25.01, *SD_females_* = 7.49, *Mean_males_* = 21.94, *SD_males_* = 7.86).

Appendix A presents the results from the linear regression models. In the subset, the only significant association observed was between the CCWS and the EREC scores, where participants with higher climate change worry also had higher eating-related eco-concern.

## 4. Discussion

The current study created and validated a screening tool to assess eating-related concerns and behaviors related to climate change. The 10-item EREC scale has shown good validity and internal consistency. First, results from factor analysis, a tool for establishing construct validity, support the single factor structure, indicating that a single latent variable fits the data well. This result is consistent with the CCWS, which was also represented well by a single factor [21]. Second, the EREC score showed a strong correlation with the CCWS score, and the Bland–Altman analysis demonstrated acceptable agreement between these two scales. This may be because both scales were designed to measure eco-concern, although the CCWS assesses eco-concern in general and the EREC scale focuses specifically on eating-related eco-concern. On average, participants scored lower on the EREC scale than the CCWS, which suggests that all individuals who worry about climate change in general do not necessarily alter their eating behaviors as a personal contribution to slowing climate change. In contrast, correlation of the EREC score with the global EDE-Q score was small and the Bland–Altman analysis demonstrated low agreement between the EREC score and EDE-Q global score, suggesting that these two measures captured distinct constructs. This is the desired outcome since the EREC scale was designed to capture a novel dimension (i.e., eating-related eco-concern) that is not assessed by the EDE-Q, as the latter represents a broader general scale of eating disorder cognitions and behaviors.

Female participants showed a significantly higher level of climate change worry than male participants, corroborating the original CCWS study [21]. Evidence on sex/gender difference in general eco-concern is well established [43]. Compared with men, women are more likely to believe that climate change is happening [44], perceive more risks of climate change [45], and have greater levels of concern about climate change [46]. Contrary to the hypothesis, we did not observe a significant sex difference in eating-related eco-concern, suggesting that although women show a higher level of concern about climate change than men, their eating behaviors do not change more due to eco-concern.

After adjusting for age, eating-related eco-concern was positively associated with climate change worry, which is understandable as individuals who are generally worried about climate change might change their eating behaviors due to eco-concern. Climate change worry was not significantly associated with the EDE-Q global score or any disordered eating characteristics, indicating that a general worry about climate change might not be directly associated with one’s level of disordered eating. Further, eating-related eco-concern was positively associated with the EDE-Q global score and shape concern while adjusting for age. One possible mechanism underlying these associations is selflessness (i.e., the tendency to relinquish one’s own needs for others’ needs), which has been associated with elevated risk of disordered eating [47]. Selflessness might also impact one’s food choices: individuals with a higher degree of selflessness are more likely to consume fruits, vegetables, and grains compared with those with a lower degree of selflessness [48,49]. Thus, selflessness may be an important mediating factor underlying the association between eating-related eco-concern and disordered eating, particularly shape concern. Indeed, previous studies have examined the association between certain types of diet, such as veganism, vegetarianism, and semi-vegetarianism, with disordered eating, yielding mixed results [50,51,52,53,54]. The inconsistencies might be due to various motivations underpinning vegetarianism, such as ethical, religious, environmental, and health concerns [52,55,56]. Thus, future studies with sufficient statistical power should examine the association between diet choices specifically due to eco-concern with disordered eating. The current findings also raise the question of whether eating-related eco-concern contributes to subsequent disordered eating, specifically shape concern, or whether individuals with existing shape concern are more likely to change eating behaviors due to eco-concern. The cross-sectional design used in this study was unable to determine the direction of the effect.

A sensitivity analysis among individuals whose EDE-Q global scores were below clinical cut-off values was also conducted. Several differences between the sensitivity analysis and analyses in the full sample were observed. First, the EREC score did not differ significantly by sex assigned at birth in the full sample. In contrast, female participants scored significantly higher on the EREC scale than male participants in the sensitivity analysis. Indeed, the mean EREC score of females did not change significantly after excluding those who exceeded the EDE-Q cut-off value, yet the mean EREC score of males was significantly lower after excluding those who exceeded EDE-Q cut-off value. This suggest that male participants who had EDE-Q scores indicative of eating disorders were more likely to have high levels of eating-related eco-concern than female participants who meet EDE-Q cut-off values. Due to limited statistical power, we were unable to compare the strength of associations between disordered eating and eating-related eco-concern in male versus female participants. Furthermore, EDE-Q global score and shape concern subscale score were significantly associated with eating-related eco-concern, yet these associations became nonsignificant after excluding those who exceeded EDE-Q cut-off values. These results suggest that the associations between eating-related eco-concern and disordered eating were more pronounced in those whose scores on the EDE-Q were indicative of possible eating disorders. These findings raise the question of direction of causality. Do individuals with eating disorders further modify their eating behaviors due to eco-concern, and/or do individuals who change their eating behaviors due to eco-concern may develop subsequent eating disorders? A longitudinal study is needed to elucidate the causality of these relationships to provide more evidence on the prevention and/or intervention of disordered eating and eating disorders.

Importantly, all questions in the EREC scale do not necessarily represent disordered eating. The goal of this study was to explore a range of behaviors that may be associated with modifications in eating behavior and attitudes related to climate change. Behaviors assessed in the EREC scale, like many healthful behaviors (e.g., eating fewer highly processed foods, reducing consumption of red meat, eating more local foods), when taken to an extreme, or when seen together with a more concerning degree of restrictive eating, could represent a component of a concerning behavioral pattern. Indeed, one of the challenges of treating individuals with eating disorders who have concerns about climate change is differentiating between truly health-damaging behaviors and behaviors that are sensible and deserve to remain in their eating repertoire because they are sustainable and climate-friendly while being compatible with recovery from an eating disorder.

In summary, this initial study developed and validated the first scale to assess eating-related eco-concern, and is the first study to examine the association between climate change worry, eating-related eco-concern, and disordered eating. Findings from this study will raise awareness of this new topic within the field of disordered eating and eating disorders. Several limitations need to be taken into consideration. First, due to limited time and funding, we only recruited one sample where participants only completed the survey once; thus, it was not possible to examine the test–retest validity using multiple samples. Second, the sample was predominately comprised of White/Caucasian participants in the United States. This limits the generalizability of the findings. Future research should validate this scale in individuals who live in different geographical or climatic regions and in a more diverse sample. Additionally, disordered eating was evaluated via a single scale. Studies using other measures and populations (i.e., individuals with eating disorders) will expand understanding of the association between eco-concern with disordered eating or eating disorders. Lastly, it was not possible to examine the temporal sequence of the associations among climate change worry, eating-related eco-concern, and disordered eating due to the cross-sectional design of this study; thus, causality could not be determined. Longitudinal studies are needed to establish a potential relationship.

## 5. Conclusions

The current study developed a 10-item scale to assess eating-related eco-concern and explored its association with disordered eating based on clinical observations in eating disorder treatment. These findings can bring awareness to the potentially new area within the fields of disordered eating, eating disorders, and climate change, which is important from both a research and clinical perspective. Healthcare providers should consider whether individuals endorse eating behaviors due to eco-concern while screening for or assessing other disordered eating behaviors. Prevention strategies, including further education about the impact of climate change and eco-concern on disordered eating, eating disorders, and mental health in general, should be incorporated into activities such as climate action clubs at schools or universities. Future studies should examine psychometric properties of the EREC scale in diverse samples, evaluate associations between eating-related eco-concern with eating disorder diagnosis and other mental health conditions, and further explore sex differences and temporal sequence of these associations.

## Figures and Tables

**Figure 1 nutrients-14-04517-f001:**
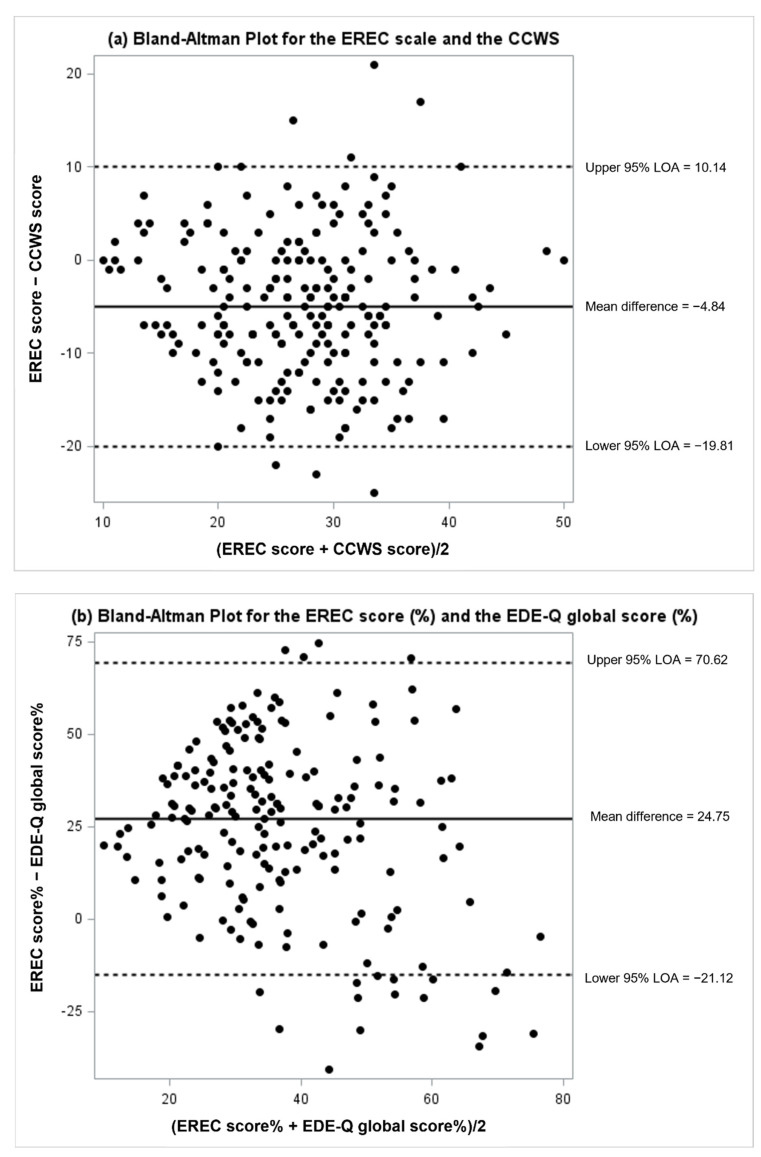
Bland-Atman plot for (**a**) the EREC scale and the CCWS and (**b**) the EREC scale (%) and the EDEQ global score (%).

**Figure 2 nutrients-14-04517-f002:**
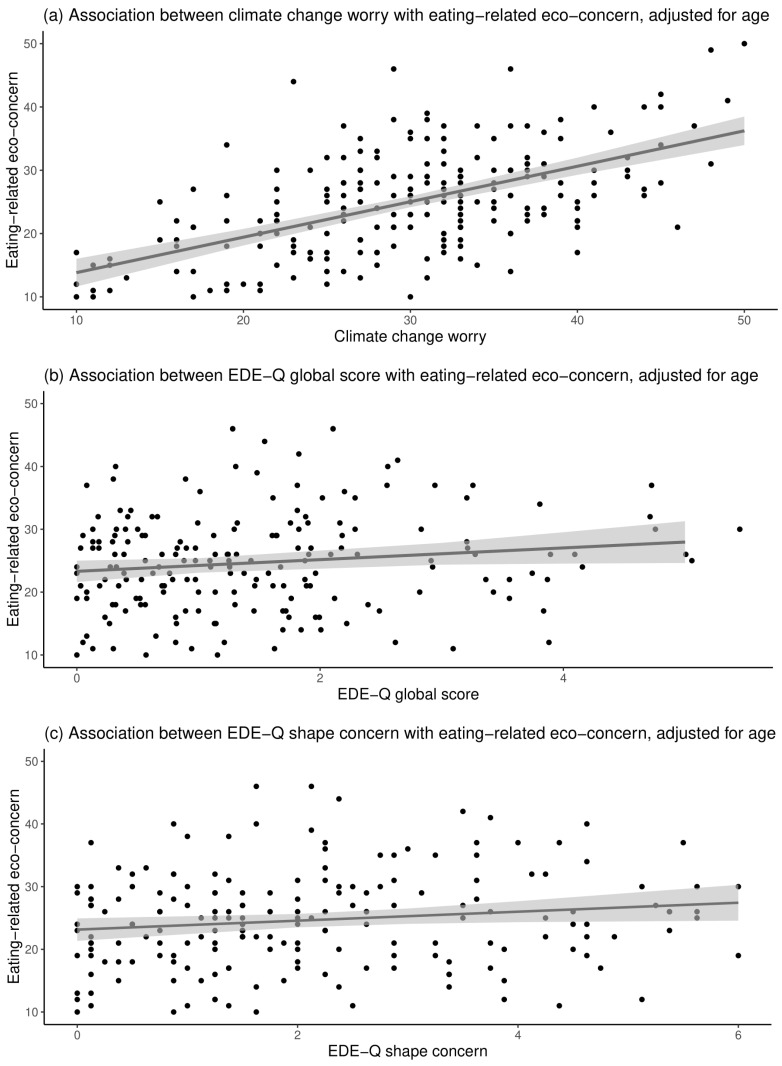
Scatter plots for the association between (**a**) climate change worry with eating-related eco-concern; (**b**) EDE-Q global score with eating-related eco-concern; and (**c**) shape concern with eating-related eco-concern. Grey area represents 95% confidence interval.

**Table 1 nutrients-14-04517-t001:** Descriptive statistics for climate change worry and eating-related eco-concern scores, and for the EDE-Q global and subscale scores.

Measure	*n*	*Mean* (*SD*)	Range
Climate change worry	224	29.72 (8.39)	10–50
Eating-related eco-concern	224	24.88 (8.03)	10–50
EDE-Q Global	193	1.48 (1.20)	0.00–5.45
Restraint	207	1.33 (1.46)	0.00–6.00
Eating concern	193	0.71 (1.11)	0.00–5.40
Weight concern	194	1.85 (1.48)	0.00–6.00
Shape concern	194	2.11 (1.57)	0.00–6.00

Note: EDE-Q = Eating Disorder Examination-Questionnaire; *SD* = standard deviation. Higher scores indicate higher levels of climate change worry, eating-related eco-concern, and disordered eating.

**Table 2 nutrients-14-04517-t002:** Descriptive statistics and factor loadings for items included in the Eating-Related Eco-Concern Scale (*n* = 224).

Item	*Mean* (*SD*)	Factor Loading
1. I spend more time than other people searching for sustainable food.	2.44 (1.06)	0.71
2. I avoid eating meat due to concerns about climate change.	2.77 (1.40)	0.72
3. I avoid eating any animal products due to my concerns about climate change.	2.03 (1.17)	0.71
4. I try not to waste food due to concerns about climate change.	3.19 (1.29)	0.63
5. I actively encourage others to change their behaviors to slow climate change.	2.63 (1.11)	0.66
6. I try to eat less because of my concerns about climate change.	1.62 (0.92)	0.59
7. I avoid genetically modified foods due to concerns about biodiversity loss.	1.85 (1.13)	0.64
8. I try to only eat organic foods or food produced without pesticides.	2.49 (1.21)	0.55
9. I avoid foods that come with excess or non-recyclable packaging.	2.97 (1.06)	0.67
10. I pay close attention to information on the impact that certain foods have on the environment (e.g., overfishing, greenhouse gasses, irrigation).	2.89 (1.08)	0.79

Note: *SD* = standard deviation. The score for each item ranges from *1* = *Never* to *5* = *Always*.

**Table 3 nutrients-14-04517-t003:** Associations among climate change worry, eating-related eco-concern, and disordered eating.

Association between Climate Change Worry and Eating-Related Eco-Concern			
*Predictor*	*β* (*SE*)	*t* (*df*)	*p*
Climate change worry	0.56 (0.05)	10.76 (1)	**<0.0001**
**Associations between Each Disordered Eating Characteristic and Climate Change Worry**			
*Predictor*	*β* (*SE*)	*t* (*df*)	*p*
EDE-Q Global	0.51 (0.48)	1.06 (1)	0.2904
Restraint	0.22 (0.39)	0.57 (1)	0.5670
Eating concern	0.65 (0.53)	1.23 (1)	0.2206
Weight concern	0.36 (0.39)	0.93 (1)	0.3535
Shape concern	0.40 (0.37)	1.09 (1)	0.2760
**Associations between Each Disordered Eating Characteristic and Eating-Related Eco-Concern**			
*Predictor*	*β* (*SE*)	*t* (*df*)	*p*
EDE-Q Global	0.93 (0.45)	2.05 (1)	**0.0414**
Restraint	0.50 (0.37)	1.36 (1)	0.1745
Eating concern	0.89 (0.50)	1.79 (1)	0.0751
Weight concern	0.71 (0.37)	1.95 (1)	0.0531
Shape concern	0.71 (0.35)	2.07 (1)	**0.0402**

Note: *df* = degree of freedom; *SE* = standard error. Age was adjusted in all models. Significant results (*p* < 0.05) are bolded.

## Data Availability

Not applicable.

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
