# Peer review of "Development and Validation of an Eating-Related Eco-Concern Questionnaire"

_nutrients, 2022, doi:10.3390/nu14214517_

Round 1
Reviewer 1 Report
Taking up an interesting, current and needed research topic - which should be congratulated on the Authors, nevertheless, the following are disturbing:
- In the validation the Authors “examined construct validity via factor analysis, convergent validity by correlating EREC score with CCWS and EDE-Q global scores, and internal consistency using Cronbach’s alpha“ without analysis test – retest in various selected population groups
- In the Abstract the phrase “Factor analysis results support a single factor structure” may be unclear for the reader (lines 30-31)
- In the Abstract the Authors wrote “Sex differences in CCWS and EREC scores were evaluated using t-tests” - the article does not provide a detailed presentation of these results
- In the Introduction, the authors should precisely define the term of eating disorders - does this concept also include anorexia and bulimia?
- The Authors wrote in the Introduction “We respond to current research gaps by: (…) 2) examining sex differences in climate change worry and eating-related eco-concern” (lines 91-92)- - the article does not provide a detailed presentation of these results (also lines 95-96)
- In the Introduction the phrase “1) the eating-related eco-concern (EREC) scale would comprise a single factor and show good validity and reliability” may be unclear for the reader (lines 94-95)
- Did the authors receive the approval of the bioethics committee to conduct the study? What does the sentence mean “The University of North Carolina at Chapel Hill Biomedical Institutional Review Board determined this study to be exempt from further review.” (lines 114-115)
- The study covered a slightly small population (n = 224), with an unequal distribution, e.g. by gender, BMI, etc; the group was not deliberately selected and is not representative of the population; in-depth detailed characterization of the group is necessary (suggestion in the table).
To sum up, the title should rather say that this is a pilot study than the target result. The article should contain clear methodological principles.
Reviewer 2 Report
The article entitled "Development and Validation of an Eating-Related Eco-Concern Questionnaire" is a very interesting read. Its focus is on examining and developing a new questionnaire, that will be useful in examining abnormal eating behaviours in a group of people suffering from mental problems related to climate change and concerns about the environmental state.
In the introduction section authors are presenting sufficient background to the studies. They explain what the most common symptoms that eco-concern people met are, and who is most at risk to suffer these consequences. They try to connect it to eating behaviours as well. In lines 73-74 they, however, start connecting eco-concern with disordered eating and in the next sentence (74-76) they point to studies (ref. 19) which show such behaviours as eliminating meat, seeking organic food, or avoiding food waste. This suggests that those behaviours can be considered disordered eating, which is not. It's just an eating behaviour which is harmless for them and it's environmentally friendly. And in the next sentence, they point to studies that are connecting warmer climates with eating disorders (bulimia nervosa). First of all in my opinion it's mostly premature to connect it to climate (which authors are pointing out later). But the most important thing is the lack of differentiation of disordered eating from eating disorder, and this is mostly ongoing in the whole article.
Disordered eating and eating disorder are not one and the same, although many authors are using them interchangeably. "Disordered eating" is general term used to describe the spectrum of abnormal and harmful eating behaviours that are used in a misguided attempt to lose weight and/or maintain a lower than normal body weight. "Eating disorder", in turn, is a term that refers to one of six clinically diagnosable conditions recognised in DSM-5, including anorexia nervosa, bulimia nervosa, BED, other specified and unspecified eating disorders and avoidant/restrictive food intake.
It is worth mentioning, that the authors used in the validation process an Eating Disorder Examination-Questionnaire, which clearly refers to eating disorders (and all references they used in section 2.2.4. are about validation, reliability and potential usefulness of this test in eating disorders like bulimia, anorexia or bed syndrome). The authors pointed out wrong in line 139 that this questionnaire is to measure disordered eating because not all disordered eating is an eating disorders. On the other hand, most of the behaviours in eating disorders are disordered eating, but not all as well, because not all are referring to eating (for example too much exercise). This needs to be clarified what the authors are referring to in the whole article because in the introduction section they are mixing all terms together, then in the whole article they are writing about disorder eating, and in the conclusion section, they are writing about eating disorders.
My second concern is about inclusion and exclusion criteria. The authors pointed out that there were no exclusion criteria in the study, and the only inclusion criteria were to be over 18 years old. My concerns are about the potential participation of people with some mental disorders. I would like to ask the authors if they are not concerned about the participation of people with eating disorders or other mental states that can interfere with the results obtained by EDE-Q?
Also, I would like to ask the authors, if they give instructions about proper weight and height measurements to participants (lines 118-120). Different times of measurements can result in huge differences in body weight so it can't be used in i comparison between people who did measurements in the morning, in the afternoon and in the evening. The weight measurements in different scales also seem to be improper, however, I understand the internet nature of questionnaires and studies and that it is a pilot study.
I did mention it before, but I want to be especially underlined - I would like to ask the authors to clarify why they are using the EDE-Q questionnaire (which is meant to be used in eating disorder studies) in the disordered eating studies.
The EREC questionnaire seems to be very useful and well constructed. The authors rightly form the question, which should be understood that potential participants very well. I would like to ask the authors if they were concerning questions about using specific diets (planetary diet, organic diet, vitarian diet, vegan diet, etc.) or using a zero-waste strategy in the case of food choice and cooking or if they are using only self grow-up vegetables (like a little garden on their own balcony or little garden on private backyard)? Eliminating some products (like animal products in question 3), eating less (question 6) is not always a special diet, not wasting food (question 4) is not zero-waste and non-pesticieds and genetically modified ecological food (questions 7 and 8) is not own grown food. Why do authors pick these particular questions? I would like to ask them to clarify the reasons for this choice in the method section. Also to clarify and give the proper studies that show that genetically modified food is prooved to participate in biodiversity loss.
I would like to ask the authors as well to clarify what test did they use to check the normality distribution of obtained data (lines 157-158).
For the validation process, the authors used Pearson correlation coefficients and they completed the comparison between EREC and EDE-Q and CCW with a linear regression model. Although seeing Pearson correlation is interesting data, and it is good to see statistical strength between those data. However, the correlation model is not very useful in the validation process, which was widely described in some literature. The correlation coefficient only measures the strength and direction of the relationship between two measurements made by different methods but does not measure their consistency. The use of a correlation coefficient can therefore result in errors in interpretation and quality assessment of the validated method. Two measurements made by different methods may show little agreement, but the calculated correlation coefficient may be high. This can be actually pretty important in the case of correlation between the EREC and EDE-Q, where p-value was 0.0455, which is on the verge of statistical significance. The Bland-Altman method should be considered by the authors to apply to perform a proper validation process of the EREC questionnaire.
I would also like to propose to the authors to present data in graphic form, especially the regression model, which will be useful in interpretation.
The discussion section is mostly properly conducted, however, it is premature to make some review before the previous issues (disordered eating vs eating disorders; validation process) would be clarified and corrected. Some results can be different after adjusting proper methods.
In conclusion, I consider this article as an interesting topic, however, it needs to be corrected before further consideration for publication in the Nutrients journal.
Reviewer 3 Report
The manuscript entitled ‘Development and Validation of an Eating-Related Eco-Concern Questionnaire’ presents interesting issue, however some corrections are needed.
Lines 49-50 – ‘However, the connection between eco-concern and disordered eating remains understudied’ – references are needed
– The EREC questionnaire should be presented in the supplementary material (in table 1 authors presented 10-items, but whole questionnaire should be presented also)
– Table 1 should be rather presented in the results section
– Line 192 – ‘Our sample…” maybe „the sample”? Scientific writing has traditionally been in third person, passive voice.
– Line 213 – correlation - The analysis of correlation is not the recommended method (so Authors should not conclude on the basis of it). At the same time, the kappa statistic and Bland-Altman method are the recommended methods. They should calculate the Bland-Altman index (in %) and conclude on the basis of the commonly indicated criteria
– Line 291 – ‘we were not able to examine the test-retest validity using multiple samples’ - Test-Retest Reliability should be also conducted
– Although the study is interesting, it is not fully validated.
Reviewer 4 Report
This is an interesting and well-written manuscript. Appropriate methodology was applied to validate a new instrument (EREC) for measuring the impact of eco-concern on eating behaviors. The topic is timely and the EREC potentially useful to many in the field.
Minor points
Participants – 224 were included but table 2 indicates not all participants completed the EDE-Q in full. Please explain the circumstances under which less than complete responses were included.
Table 2 – suggest providing a footnote to remind readers about the scoring, ie. high score = high concern, etc. (as sometimes the opposite is true)
Line 200 – suggest reducing the number of significant figures for BMI as it’s unlikely self-report would yield such precision
Round 2
Reviewer 1 Report
I would like to thank the Authors for very careful and extensive supplementation of the content of the article, as well as comprehensive answers to suggestions and questions (including the consent of the bioethics committee). I accept the text as it stands.
Author Response
We thank the reviewer for their careful review.
Reviewer 3 Report
I appreciate the great efforts that the authors have made in response to my questions and concerns. The level of agreement between the EREC with the CCWS scores are moderate (seems to be acceptable) but between the EREC and EDE-Q not good. Authors should calculate the Bland-Altman index (in %) and conclude on the basis of the commonly indicated criteria (e.g. presented by Myles & Cui, https://www.ncbi.nlm.nih.gov/pubmed/17702826).
It is important study but the limitation must be clearly presented.
